# Research Progress of Nanomaterials Acting on NK Cells in Tumor Immunotherapy and Imaging

**DOI:** 10.3390/biology13030153

**Published:** 2024-02-27

**Authors:** Yachan Feng, Haojie Zhang, Jiangtao Shao, Chao Du, Xiaolei Zhou, Xueling Guo, Yingze Wang

**Affiliations:** College of Food Science and Biology, Hebei University of Science and Technology, Shijiazhuang 050018, China; feng15630469392@163.com (Y.F.); zhj15630338281@163.com (H.Z.); sjt0903@126.com (J.S.); duchaopku@126.com (C.D.); foxlei@live.cn (X.Z.)

**Keywords:** NK cell immunotherapy, nanomaterials, tumor immunotherapy, target block, drug carrier

## Abstract

**Simple Summary:**

The development of novel cancer treatment techniques is essential because malignant tumors represent a severe threat to human life and health. By inhibiting immunological sites and chimeric antigen receptors, tumor immunotherapy is becoming more and more significant. There are several immune cells linked to tumors, and NK cells are one type that is being utilized more and more in therapeutic settings because of the inherent benefits. Additionally, the anti-tumor immune response and NK cell-tracking ability are improved by the use of nanomaterials. In order to accomplish the combined application of NK cells and nanomaterials, this review will concentrate on the immunological application and imaging study of nanoparticles on NK cells.

**Abstract:**

The prognosis for cancer patients has declined dramatically in recent years due to the challenges in treating malignant tumors. Tumor immunotherapy, which includes immune target inhibition and chimeric antigen receptor cell treatment, is currently evolving quickly. Among them, natural killer (NK) cells are gradually becoming another preferred cell immunotherapy after T cell immunotherapy due to their unique killing effects in innate and adaptive immunity. NK cell therapy has shown encouraging outcomes in clinical studies; however, there are still some problems, including limited efficacy in solid tumors, inadequate NK cell penetration, and expensive treatment expenses. Noteworthy benefits of nanomaterials include their chemical specificity, biocompatibility, and ease of manufacturing; these make them promising instruments for enhancing NK cell anti-tumor immune responses. Nanomaterials can promote NK cell homing and infiltration, participate in NK cell modification and non-invasive cell tracking and imaging modes, and greatly increase the effectiveness of NK cell immunotherapy. The introduction of NK cell-based immunotherapy research and a more detailed discussion of nanomaterial research in NK cell-based immunotherapy and molecular imaging will be the main topics of this review.

## 1. Introduction

Being the second-largest cause of mortality worldwide, cancer frequently has a rapid rise in patient numbers along with a steady rise in treatment complexity. It is also starting to rank among the world’s most pressing issues that need to be resolved [1]. Chemotherapy, radiation therapy, and surgery are examples of classic cancer treatments that are now producing positive clinical outcomes. However, traditional treatment is often accompanied by clinical symptom specificity, low tumor response rate, and patient resistance. In order to avoid the issues and restrictions associated with conventional treatment approaches, novel approaches to cancer treatment are still required.

James P. Allison, a US immunologist, and Tasuku Honjo, a Japanese immunologist, were given the 2018 Nobel Prize in Physiology or Medicine for their discoveries of cancer treatments that block negative immune regulation, which have successfully accelerated the development of tumor immunotherapy [2,3,4]. Tumor immunotherapy has made great strides in treating malignant tumors in recent years. Among the many challenges it faced were immune checkpoint blocking (ICB), chimeric antigen receptor (CAR) cell therapy, and tumor vaccines. Among them, tumor immunotherapy based on natural killer cells is also emerging as a therapeutically viable therapeutic approach [5]. As innate immune cells, NK cells primarily depend on the equilibrium of cell surface signals to preserve the stability of patients’ tissues. They also employ their own cytotoxicity to track down and eliminate malignant cells. Compared to T cells, which require particular antigen activation, this means that the clinical application prospects of NK cell-based anti-tumor therapies are comparatively wider [6].

Nevertheless, there are issues with NK cell-based immunotherapy as well, including low NK cell supply, delayed tumor immunosuppression, and decreased cell survival and homing. However, because of their own efficient load transmission, interactions with immune cells, and material structure change, nanomaterials have successfully advanced NK cell immunotherapy [7,8,9]. Furthermore, there is a lot of promise for novel immunotherapy due to the adaptability of nanomaterials and the suitability of NK cells.

This review clarifies the application prospects and limitations of NK cell-based immunotherapy and introduces the research progress and application status of nanomaterials in accordance with their limitations. It also focuses on the research progress of NK cell-based immunotherapy from the perspective of tumor immunotherapy methods and research status. In order to advance technology and raise the effectiveness of tumor immunotherapy, the particular mechanisms and clinical benefits of nanomaterials acting on NK cells in tumor immunotherapy are finally compile.

## 2. NK Cell Research Advancements in Immunotherapy and Imaging

The progress of human health has faced significant challenges due to the advent of malignant tumors. Nowadays, chemoradiotherapy, surgery, and other treatments are the most often utilized cancer therapies. Still, patient resistance, clinical heterogeneity, and tumor metastasis contribute to the limited effectiveness of conventional treatments. As science and technology have advanced in recent years, tumor immunotherapy has surfaced as necessary. In 1893, American surgeon William Coley made the groundbreaking discovery that bacteria might stop sarcomas from growing. This marked the first instance in the history of human tumor immunosuppression [10]. Tumor immunotherapy was proposed in the 1980s and 1990s after the relationship between immune cells and melanoma was identified [11,12]. Recent clinical trials have demonstrated that tumor immunotherapy can considerably increase patients’ quality of life in addition to extending their survival time [13]. Among these, tumor immunotherapy, which uses T cells as the primary agent, has the potential to greatly expand patient treatment options while also enhancing clinical treatment outcomes. T cells provide a good starting point for tumor immunotherapy, and in recent years, more and more tumor-related immune cells have played a crucial role in tumor treatment.

### 2.1. NK Cells

Tumor-promoting immune cells and anti-tumor immune cells are the two main groups of cells linked to cancer immunotherapy. These immune cells collaborate in tumor immunotherapy and cause mutual interference (Figure 1). NK cells are some of the immune cells that fight tumors. They primarily make use of the “self deficiency” mechanism to stop self-inhibitory receptors from attaching to MHC-1 on the tumor surface, which activates the self-transmission of anti-tumor immune signals.

#### 2.1.1. Origin of NK Cells

Peripheral blood (PB), umbilical cord blood (UCB), bone marrow (BM), human embryonic stem cells (hESCs), and generated NK cell lines are the main sources of NK cells employed in tumor immunotherapy. The benefits and drawbacks of NK cells derived from various sources vary (Table 1) [14].

Before being administered, NK cells must go through an in vitro amplification and activation procedure. The objective of NK cell multiplication in vitro for adoptive metastasis is to augment endogenous cytotoxicity and re-enter the tumor site while preserving self-defense. In order to facilitate cell expansion and maturation, culturing in an appropriate medium—which may be enhanced with serum or by adding cytokines (IL-2, IL-15), antibodies, and other small molecules—is the first step in achieving cell expansion (differentiation). Typically, the amplification plan lasts two to three weeks. After obtaining the required number of cells with the proper phenotype, NK cells are gathered in the last reagent and promptly activated for additional therapy prior to infusion into the patient’s body.

It has been demonstrated that NK cells, whether autologous or allogeneic, are safe and well tolerated in therapeutic settings. For example, in Marin D et al.’s study, 37 patients with CD19 B-cell malignant tumors participated in a phase 1/2 experiment using natural killer (NK) cells generated from umbilical cord blood that expressed anti-CD19 chimeric antigen receptor and interleukin-15 (CAR19/IL-15). According to research, autologous CAR19 T cells and CAR19/IL-15 CBU-NK cells both show comparable therapeutic efficacy. However, the safety features are considerably different because CAR19/IL-15 CBU-NK cells are not significantly linked to CRS or neurotoxicity. Additionally, the improved anti-tumor activity of CAR/IL-15 NK cells from the ideal CBU in vivo has been verified using a variety of mouse models. The study’s conclusions highlight how crucial it is to establish criteria for allogeneic donor selection in order to produce CAR-NK cells and identify predictors of donor-specific reactions [15]. Despite the positive aspects of in vitro collection, many obstacles persist that prevent its continued advancement. Allogeneic NK cells are hard to come by, and autologous NK cells have comparatively low anti-tumor efficacy and low proliferation. Furthermore, it takes time and money to separate NK cells from peripheral blood since they can combine with monocytes and other blood cells. Therefore, new technical tools are required to help apply NK cell therapy more effectively.

#### 2.1.2. NK Cell Classification and Subsets

NK cells can be divided into CD56(bright) and CD56(dim) according to the expression of CD56 on their surfaces [16]. Among them, CD56(bright) NK cells mainly play an anti-tumor immune role by secreting cytokines such as INF-γ and TNF-β [17]. CD56(dim) NK cells use highly expressed CD16 to mediate antibody-dependent cell-mediated cytotoxicity (ADCC) to induce phosphorylation of immune receptor tyrosine activation motif (ITAM) and finally immunize and kill tumor cells [18].

Using high-dimensional single-cell RNAseq, Crinier et al. have recently demonstrated the heterogeneity of human and mouse NK cells. They have also verified that NK cells can be further subdivided into multiple cell subsets based on the degree of differentiation, including cytotoxic NK (cNK) cells, antigen-presenting NK (AP-NK) cells, helper NK (NKh) cells, and regulatory NK (NKreg) cells [19]. Many NK cell subsets have been extensively employed in clinical settings and have diverse functions in various domains of anti-tumor immunity.

#### 2.1.3. Role and Function of NK Cells

Being the immune system’s first responders, NK cells are essential for immunological control, surveillance, and anti-tumor immunity (Figure 2).

NK cells can directly cause autotoxicity by secreting granzyme and perforin. Second, NK cells express high levels of CD16 to initiate ADCC when they encounter a stimulatory signal that is greater than an inhibitory one. Furthermore, NK cells have the ability to release cytokines and chemokines that attract and assemble other immune cells (T cells, DC cells, etc.), initiating an adaptive immune response that destroys tumor cells [20,21,22]. More significantly, previous sensitization and specific antigen presentation are not necessary for NK cell activation. As a result, NK cells have the ability to both trigger a systemic tumor immune response and specifically target tumor cells.

But in order for NK cells to perform their function, they must be stimulated. NK cells are currently understood to control the dynamic equilibrium between signals transmitted by a variety of inhibitory and activated receptors. Studies have indicated that the following signal molecules are mostly capable of influencing NK cell activity: MHC-1, IL-2, IL-15, IL-18, IL-12, and IL-21. Among these, the overexpression of activated receptor ligands or the lack of MHC-1 inhibitory receptor ligands can activate natural killer cells and set off immunological responses against them. Subsequent research showed that normal cell surfaces can prevent NK cell activation by expressing MHC-1 molecules, which cause inhibitory signal transduction in NK cells. However, by downregulating MHC-1 expression, tumor cells or viral infections can avoid being recognized by CD8^+^ T cells. A significant amount of granzyme and perforin are released by activated NK cells to cause target cell death. By encouraging human peripheral monocytes to increase cytotoxic cells, specifically lymphocyte-activated killer (LAK) cells, IL-2 can directly stimulate NK cell proliferation and activation to disrupt tumor cell development. The most promising substitute for IL-2 is IL-15, which can increase the activity of TME-depleted NK cells, repair their mitochondrial integrity, upregulate granzyme B production, and ultimately increase IFN-γ and NK cell toxicity. The combination of IL-18 and IL-12 induces the expression of FASL in NK cells, and by enhancing the production of IFN-γ and TNF in NK cells, inducing the expression of CCL21 and CCR7, thereby improving NK cell activity. By upregulating NKp30 and expressing perforin enzyme B, IL-21 primarily increases the activation and proliferation of natural killer cells (NK cells) while also augmenting their cytotoxic potential. For instance, NK cell treatment based on tetraspecific antibodies was developed by Demaria et al. Studies have revealed that signaling molecules, like IL-2, can promote human NK cell activation and proliferation [23]. Furthermore, it has been shown by Shoubao Ma et al. that IL-15 can markedly improve the ability of NK cells to survive, persist, and activate in vivo [24].

### 2.2. Immunotherapy Based on NK Cells

#### 2.2.1. Target Blockade

The primary strategy for shielding host cells from foreign antigen harm is to maintain a balance of immunological signals, as immune targets are essential for immune homeostasis and host survival. Currently, hematological malignancies and solid tumors are frequently treated with ICB, and a growing number of targets for tumor immunotherapy have been identified (Figure 3). More crucially, studies have shown that the balance between the inhibitory and activated receptors on the surface of natural killer cells (NK cells) governs their immunological function. Tumor cells will be encouraged to evade the immune system when the inhibitory signal is higher than the activation signal. Therefore, by identifying and obstructing relevant inhibitory signals, NK cells’ anti-tumor immune response can be enhanced. PD-1, TIGIT, and others are the most commonly employed inhibitory targets.

As an inhibitory receptor, PD-1 primarily blocks T cells’ ability to mount an immunological response on behalf of the patient [25]. PD-L1 and PD-L2, which are extensively expressed on the surface of tumor cells, are the principal binding ligands for it. Its particular mode of action involves using ligand binding to mediate immunological escape from tumor cells, suppress T cell activation, and send negative co-stimulatory signals [26,27]. Furthermore, clinical studies show that PD-1 is highly expressed in the pleural effusion of patients with primary and metastatic malignancies in addition to immune cells like NK cells [28]. It is important to note that PD-1 influences NK cell-reliant immune monitoring and encourages tumor cell immune escape in addition to mediating NK cell inactivation and downregulating its anti-tumor immune response [29,30]. Given that this suggests that PD-1 may systematically regulate NK cells’ anti-tumor immune response, targeted PD-1 inhibition is crucial for enhancing the therapeutic efficacy of NK cell-based tumor immunotherapy [31].

NK cells exhibit high levels of expression for TIGIT, a type I transmembrane protein belonging to the immunoglobulin superfamily. Its primary mode of action is binding to ligands produced on antigen-presenting cells, such as CD155 and CD112, which then activates NK cells and sends anti-tumor immunological signals [32,33,34]. Of them, CD155 is the primary binding ligand and is highly expressed on the surface of tumor cells [35]. It was also found that blocking TIGIT signaling pathway of malignant tumors could significantly increase the expression of IFN-γ and TNF-α in tumor-specific CD8+ T cells and significantly improve the anti-tumor immune response [36]. Furthermore, TIGIT binds to ligands for self-phosphorylation and inhibits the PI3K/MAPK signaling pathway, which in turn lessens NK cell cytotoxicity [37]. As a result, TIGIT monoclonal antibodies can be employed to achieve therapeutic objectives by preventing the spread of immunosuppressive signals, boosting NK cells’ anti-tumor activity, and controlling patients’ immune systems.

Target blockade is a common tool in tumor immunotherapy; however, prolonged exposure to the tumor microenvironment alters the molecular expression of NK cell-activated receptors, which reduces the ability of NK cell-mediated immune surveillance. Consequently, to increase the therapeutic benefit of tumor immunotherapy, new treatments must either appear or work in tandem with existing ones.

#### 2.2.2. CAR-NK

Tumor immunotherapy has been using CAR-T therapy more and more in recent years. Tumor immunotherapy is based on the augmentation of immune lymphocytes; however, the incidence of anti-tumor immune responses is strictly restricted due to the absence of endogenous T cells [38,39]. In order to express antigens that precisely target tumor cells, T lymphocytes from patients (autologous) or healthy people (allogeneic) were gathered and genetically modified. Ultimately, the patient receives the effectively altered T lymphocytes in order to produce immune responses specific to the tumor and eliminate the tumor cells. Chimeric antigen receptor (CAR) therapy is the collective term for this immune cell-based anti-tumor therapy (Figure 4) [40].

CAR-NK was developed in response to the strong clinical results of CAR-T, and it has since undergone testing and optimization in both animal models and clinical studies. For instance, TME has a significant impact on the therapeutic use of CAR-NK in the treatment of solid malignancies. Hypoxia, an acidic or high-adenosine environment, NK cell surface receptors, and TGF-β can all have an impact on CAR-NK’s anti-tumor effectiveness. By modifying intrinsic pathways, battling tumor heterogeneity, opposing inhibitory TME, and boosting CAR-NK cell infiltration into tumor tissue, it can prevent tumor cell immune escape. The mechanisms behind CAR-NK’s effectiveness in solid tumors remain unclear despite ongoing clinical trials, especially with liver cancer, cholangiocarcinoma, and urinary system cancers [41]. It was discovered that CAR-NK had both the distinct anti-tumor immunological advantage and the innate anti-tumor characteristics of NK cells (Figure 5) [42].

Three components make up the majority of the functional CAR molecules expressed on NK cells, according to the structural design of CAR-T: the extracellular domain, transmembrane domain, and intracellular signaling domain. Among these, the hinge region can establish a direct connection between the extracellular domain and the transmembrane and intracellular domains. The extracellular domain is made up of signal peptides and single-chain antibody fragments that identify antigens. The hinge area of CAR-NK mostly consists of DAP12, CD8α, CD28, and the IgG Fc domain. The transmembrane domain is an important site used to connect the extracellular and intracellular domains of CAR. The most commonly used transmembrane domains (receptors) of CAR-NK at present are CD8, CD28, and CD3ζ, and there are also highly promising transmembrane domains such as 2B4, NKp46, NKp44, NKp30, NKG2D, and DNAM-1. The intracellular domain mainly binds to CAR-NK activation signals, and the number of signal binding determines the maturity of CAR-NK. Research has found that if only CD3 ζ When combined, the first generation CAR-NK will be formed. One or two additional co-stimulatory signals, such as those from the CD28 family (CD28 and ICOS), TNFR family (CD27 and OX40), or SLAM (2B4), are also carried by the second- and third-generation CAR-NK. Therefore, by assembling the aforementioned three components, CAR-NK can effectively induce the production of the NK cell activation signal and give NK cells new capabilities for immune destruction. Furthermore, CAR-NK offers a safer clinical benefit than CAR-T. Allogeneic NK cell infusion, for instance, has been demonstrated in certain phase I/II trials to be well tolerated and to not result in GVHD or substantial toxicity.

CAR-NK therapy has been shown to have an immune effect on a variety of hematologic malignancies, including but not limited to acute myeloid leukemia, lymphoma, and multiple myeloma. It has been widely used in the treatment of CD19^+^ B cell malignancies. In order to combat CD19, Romanski et al. employed CAR to transduce human malignant non-Hodgkin’s lymphoma patients’ natural killer cells, NK-92, and they were able to show that CAR-NK-92 can preferentially lyse B cells that express CD19 [43]. Furthermore, studies demonstrate that NK cells treated with enhanced green fluorescent protein (EGFP) can both show a sustained anticancer impact in PDX mouse models and greatly increase their lytic capacity. TNBC cells with increased EGFR expression stimulated both EGFR-CAR NK cell types and selectively caused the TNBC cells to lyse in vitro [44]. As a result, triple-negative breast cancer is frequently treated with EGFP-CAR-NK [45].

After chemotherapy, radiation therapy, and surgery, tumor immunotherapy has emerged as the “fourth bullet” in anti-tumor therapy. It appears that the patient’s immune system can be “reprogrammed” to fight against malignant tumors thanks to the therapeutic use of CAR-T. A major application of antigen presentation is the use of ICBs. Consequently, immunotherapy is emerging as a novel therapeutic hotspot that can decrease the number of tumors and appropriate cell populations while also enhancing treatment efficacy through mutual benefit. But as already noted, there are numerous restrictions and a rather low application efficiency for single immunotherapy. For instance, fewer antigens specific to the tumor are present when CAR-T is utilized to treat solid tumors. As a result, there are insufficiently strong targets to restrict the choice of drugs. Moreover, a single mode of action, significant side responses, and high drug resistance are some of the drawbacks of mono immunotherapy. The one that affects patients the most among them is toxicity control. It also results in adverse events like neutropenia (19.6%), hypertension (9.3%), and lymphopenia (10.3%), in addition to clinical symptoms like anemia (45.4%), weariness (34.3%), and difficulty swallowing (30%) [46]. Presently, a growing number of immunotherapy combinations are being used in clinical settings and are eventually evolving into brand-new therapeutic approaches.

#### 2.2.3. CAR-NK Combined with Gene Modification

NK cells’ immunological surveillance can be circumvented by tumor cells. To ensure that NK cells can destroy tumor cells directly before they escape, gene modification is required to change NK cells to increase their anti-tumor immune function. Numerous genetic modification techniques have been used to modify NK cells thus far. One key way to increase the safety of CAR-NK cell treatment is to include suicide genes. Phase I clinical studies utilizing GD2-CAR-T cells with suicide genes (NCT01822652) are now being conducted to treat neuroblastoma and sarcoma [47]. Additionally, CAR-NK suicide system research is progressing daily. Among these, hematopoietic stem cell transplantation was the first application of the HSV-TK/GCV suicide system in the 1990s. Furthermore, the suicide mechanism of CAR-NK has also made use of apoptosis pathways like FAS and caspase-9. For instance, when Liu et al. transfected retroviral vectors containing *CAR-CD19*, *IL-15*, and *caspase-9* suicide genes into CB-derived NK cells, the results demonstrated a significant prolongation of the survival of Raji lymphoma mice in vivo in addition to significant cytotoxicity against *CD19* cell lines and primary leukemia at the in vitro level. Furthermore, this gene can modify the CAR-NK gene in a way that dramatically reduces the cytotoxicity of NK cells, which greatly expands the clinical therapy’s applicability constraints [48].

#### 2.2.4. Combination of Target Blocking and Monoclonal Antibody

A monoclonal antibody can specifically target the inhibitor receptor on the surface of natural killer cells, obstructing the transmission of its signals and enhancing the cells’ ability to fight tumors. NK cell surface receptors are currently the target of a large number of monoclonal antibodies used in clinical treatment (Figure 6).

Monoclonal antibodies that are blocked by distinct targets are targeted at different types of tumors. For example, anti-PD-1 primarily targets cancers of the cervical, breast, liver, and lung tissues [49]; anti-NKG2A primarily affects natural killer cells in hematological malignant tumors [50]. According to reports, the use of monoclonal antibodies that obstruct PD-l/PD-L1 binding can greatly increase NK cells’ cytotoxicity and their ability to produce immunosuppressive cytokines, which in turn stunts the growth and spread of tumor cells. Many monoclonal antibodies that target PD-1 are currently being employed in clinical settings, and promising research outcomes have been attained. For instance, the receptor for PD-1 was overexpressed in tumor cells when it was first found on T lymphocytes. Consequently, PD-1 has a major impact on how NK cells respond immunologically to activators. Investigations centered around PD-1 can elucidate its possible function in NK cell immunosuppression and anti-tumor tactics aimed at reinstating NK cell toxicity [51].

### 2.3. Limitations of NK Cell Therapy

Tumor microenvironment (TME) immunosuppression has the ability to reduce NK cells’ surface-activated receptor expression as well as prevent their activation and growth [52]. Furthermore, the production of adhesion factors can be inhibited by vascular endothelial growth factor (VEGF) and basic fibroblast growth factor (bFGF) released by tumor cells, which can impact NK cell infiltration and homing [53]. More significantly, NK cell-based tumor immunotherapy frequently has poor NK cell activity, inadequate homing infiltrate, and infrequent interaction with tumor cells [54]. Consequently, further technical methods are still required to support and lessen the application restrictions of tumor immunotherapy, even though its discovery and practical application have accelerated the advancement of cancer treatment.

## 3. Research Progress on Nanomaterials Acting on NK Cells

It is important to note that the development of nanomaterials has improved tumor immunotherapy and is progressively being used in other immunotherapy applications (Figure 7). More opportunities for tumor immunotherapy applications have been made possible by the development and sensible design of nanomaterials. Thus, there is a significant deal of scientific significance about the role of nanomaterials in NK cells’ anti-tumor immunological function.

### 3.1. Types of Nanomaterials

#### 3.1.1. Metal Nanoparticles

The biomedical area makes extensive use of metal nanoparticles [55], which also play a vital role in controlling host defense and immune cell activation [56,57]. Because of their low toxicity, chemical inertness, and ability to control light, gold nanoparticles are a popular choice for cancer patient detection and treatment [58,59,60,61]. Ahn, for instance, developed an anti-tumor vaccine using Au NPs containing an endogenous EDB autoantigen. Furthermore, studies have shown that Au NPs can stimulate T cell anti-tumor activity and encourage DC cell antigen presentation, which prevents the growth and formation of malignancies [62].

#### 3.1.2. Liposomes

The existence of liposomes was established in 1965 by Bangham et al. [63]. They discovered that liposomes are spherical, double-layered vesicles that possess both hydrophobic and hydrophilic properties. Furthermore, Chen and colleagues altered liposomes by adding anti-PD-1 and mannose, and they also enclosed anti-angiogenic medications and mTOR inhibitors within liposomes. Studies have demonstrated that this liposome has the ability to not only enhance the volume of mouse colon cancer tumors while concurrently suppressing angiogenesis and glycolysis but also rewire immune cells to enhance the efficacy of conventional anti-PD-1 therapy [64]. Since liposomes do not exhibit anti-tumor properties, they can only be utilized as a drug delivery mechanism in tumor immunotherapy, despite their great encapsulation efficiency and ease of manufacture. However, the emergence of multidisciplinary intersection gives liposomes new capabilities including targeting and immunotherapy, improving their use in clinical tumor treatment.

#### 3.1.3. Hydrogels

Because hydrogels can load therapeutic medicines efficiently and have strong biocompatibility, tumor immunotherapy frequently uses them in its design. While conventional immunotherapy works well for treating primary tumors, it frequently falls short when it comes to treating metastatic and recurring malignancies. To stop tumor spread and lower tumor recurrence, a new therapeutic strategy must be created immediately. For instance, Chen et al. created novel nanomaterials for tumor immunotherapy using PPP (PLGA-PEG-PLGA) and the ROCKs inhibitor Y27632. Due to the temperature-responsive nature of the two-component combination, the hydrogel state can be preserved during the ensuing treatment procedure. The results showed that when hydrogel entered tumor cells, it caused the cells to split into fragments and released Y27632, which made dendritic cells phagocytize the pieces and present antigens. This, in turn, activated T cells, boosting the immune response against mouse melanoma [65].

### 3.2. Application of Nanomaterials Targeting NK Cells

#### 3.2.1. Nanomaterials for Molecular Imaging of NK Cells

Clinical and contemporary research always employ immunohistochemistry biopsies to provide real-time monitoring of NK cell responses in the body, which complicates the regulation of ACT. Contrast agent-mediated molecular imaging techniques, including immunopositron emission tomography (PET), immunomagnetic resonance imaging (MRI), immunophotoacoustic imaging (PAI), etc., can be used to anticipate the NK cell response and activity in ACT. Thus, developing multimodal imaging nanoparticles for in vivo real-time NK cell response monitoring is crucial for future controllable ACT. Quantitative dynamic footprinting (qDF), total internal reflection fluorescence (TIRF) microscopy, optical live-cell imaging (including multiphoton and confocal imaging), light-sheet microscopy, super-resolution microscopy, and other techniques have also improved in their ability to assess the tracking of NK cells. Many of these techniques rely on directly labeling the surface of NK cells with fluorophores or contrast agents, cell-permeable fluorophores, radioisotopes, or other substances to enable real-time visualization of NK cell immunotherapies in tumors. NK cells have also been labeled for MRI using nanoparticles, such as ultra-small and superparamagnetic iron oxide nanoparticles. It is quite simple to introduce iron oxide nanoparticles to NK cells; either transfection agents or a straightforward incubation electroporation are needed. NK cells tagged with iron oxide exhibit a robust low-intensity signal in T2- and T2*-weighted imaging. The duration of iron oxide labeling can last for up to 4 days, the duration of labeling often depends on the length of time that the NK cells under-going adoptive transferred have survived [66].

For instance, Dong Hyun Kim et al. created magnetic nanocomposites (HAPF) that can be used to identify NK cells using protamine, hyaluronic acid, and superparamagnetic nano iron oxide—materials that the FDA has approved for clinical use. Effective adhesion of the produced HAPF to NK cells (HAPF-NK) is observed. Applying an external magnetic field stimulates natural killer cells (NK cells) and encourages the production and release of lysosomes. Magnetically activated HAPF-NK cells also allow MR imaging to guide NK cells through intraductal arterial (IA) infusion for the treatment of hepatocellular carcinoma (HCC) solid tumors. Tumor development was decreased following therapy with magnetically activated NK cells injected by IA, suggesting an increased therapeutic efficacy of image-guided local delivery of magnetically activated HAPF-NK cells. In conclusion, our work created a magnetic nanocomposite that uses magnetic field stimulation to increase the effectiveness of NK cell therapy for solid tumors and precisely track the dispersion of NK cells using MRI [67].

#### 3.2.2. Nanomaterials Enhance the Anti-Tumor Activity of NK Cells

The primary variables influencing the immunological activity of natural killer (NK) cells are the negative regulatory factors released by tumor cells, including TGF-β and IFN-γ. By reducing the expression of NK cell surface-activating receptors, they suppress activation signaling and NK cell cytotoxicity as well as the anti-tumor immune response. Therefore, inhibiting TGF-β signal transduction is a good place to start if we wish to increase the anti-tumor activity of NK cells. Liu et al., for instance, created nanoemulsions with selenocysteine and a TGF-β inhibitor. Research revealed that the nanoemulsion up-regulated the expression of the NKG2DL receptor and dramatically blocked TGF-β/TGF-βR1/Smad2/3 signaling, thereby successfully increasing the anti-tumor activity of NK cells [68]. Furthermore, low response rate, medication resistance, and patient heterogeneity are issues that nanomaterials can solve for traditional therapy to increase efficacy [69]. For instance, selenium-based nanoparticles can boost the anticancer activity of natural killer cells (NK cells) while simultaneously eliciting a non-specific immune response, hence increasing the overall immunotherapy efficacy [70,71]. Consequently, adding nanomaterials can greatly enhance the therapeutic benefit of NK cell treatment while also lowering its clinical adverse effects. Research on how cancer patients are treated is very important.

#### 3.2.3. Immune Modification of NK Cells by Nanomaterials

As the first effectors to identify and track tumor cells, NK cells are increasingly giving way to CAR-NK as the next wave of cutting-edge immunotherapy tools. Simultaneously, the advancement of CAR-NK immunotherapy has facilitated the delivery of nanomaterial-based chimeric antigen receptor genes. Through the use of their own permeability, retention effect, and aggregation energy, chitosan nanoparticles loaded with IL-21 and NKG2D genes have been shown in studies to more successfully activate NK cells in vitro and exhibit superior anti-tumor effects [72,73,74]. Therefore, by altering NK cells with pertinent cytokines, nanomaterials can dramatically suppress the proliferation of tumor cells.

Furthermore, the nanomaterial-modified synthetic natural killer cells have the ability to kill tumor cells directly. For instance, it was discovered that a new nanomaterial made of NK cells transfected with the human ferritin heavy chain (hFTH1) gene and embedded in gold nanoparticles could direct NK cells into TME and give excellent transfected NK cell imaging [75]. Consequently, by exploiting the immunological modification of NK cells, biocompatible multifunctional nanomaterials can enable the real-time monitoring of NK cells in patients [76].

#### 3.2.4. Nanomaterials Enhance NK Cell Homing and Infiltration

The homing behavior of NK cells is mainly dependent on the signal transduction between the homing receptor on the surface and the ligand, while the infiltration of NK cells is influenced by the interaction with TME [77,78]. Studies have found that once NK cells infiltrate tumor cells and homing receptors bind to ligands, immune cell activation signals will be transmitted and secrete perforin, granzyme, and apoptosis-inducing factors [79] to play the anti-tumor immune role of NK cells. In order to prevent the growth of tumors, it is therefore essential to employ nanomaterials to improve NK cell homing and infiltration. For instance, conjugating iron oxide nanoparticles to primary NK cells dramatically improved their homing abilities and increased granzyme and perforin production [80]. Furthermore, recent research has shown that external magnetic guidance also influences the homing and infiltration of NK cells. Wu et al., for instance, subcutaneously implanted polydopamine-coated magnetic iron oxide nanoparticles into mice. The outcomes demonstrated that the magnetic particles might enhance NK cell homing and infiltration while simultaneously activating NK cells to attack tumor cells directly, exhibiting more potent anti-tumor activity [81]. Tumor invasion and metastasis can be controlled by NK cells’ homing behavior. Because they affect NK cell homing, nanomaterials thus have a bigger impact on the clinical therapy of malignancies.

#### 3.2.5. NK Cell-Associated RNAi Loaded on Nanomaterials

RNA effectors, such as siRNA, miRNA, and shRNA, have the ability to silence particular immune cell genes, altering the function of their genomes and boosting anticancer activity [82]. These RNA effectors can be better delivered by using nanomaterials, which can also serve as a great nano-delivery mechanism. For instance, *CD47* and *PD-1* expression might be markedly reduced by cationic liposomes loaded with epithelial cell adhesion molecules including siCD47 and siPD-1. Furthermore, the liposome has the ability to speed up the transmission of anti-tumor immune response signals and enhance NK cell proliferation in addition to potently inhibiting tumor growth and lung metastasis. In a lung metastasis model, systemic treatment of LPP-P4-Ep may dramatically suppress the formation of solid tumors in subcutaneous mice and diminish lung metastasis in mice by effectively silencing *CD47* and *PD-L1* compared to single-gene silencing in vivo. Target delivery of LPP-P4-Ep enhanced the release of many cytokines, including IFN-γ and IL-6, and enhanced anti-tumor T cell and NK cell responses both in vivo and in vitro [83].

## 4. Conclusions

Thus, in the interaction between the immune system and tumors, the multifunctionality of nanoparticles will enable the synergistic binding of anticancer effects, expand the potential of NK cell cancer immunotherapy, and aid in the development of safe and controllable NK cell cancer immunotherapy, thereby offering more efficient clinical treatment approaches for tumor patients.

## 5. Future Directions

Tumor immunotherapy offers new hope for the prevention and treatment of malignant tumors because the human body’s immune system can defend against a variety of diseases, including cancer. However, the clinical outcomes of immunotherapy have not been satisfactory yet, and only a tiny percentage of patients respond clinically to monotherapy for tumor treatment. Furthermore, monotherapy frequently results in indiscriminate cytotoxicity, inhibition of the tumor microenvironment, and immune cell depletion. As a result, the combination of various immunotherapies has been employed extensively in recent years and has shown previously unheard-of clinical success.

Among them, tumor immunotherapy based on NK cells has attracted great attention. Tumor-specific cytotoxicity is a feature of NK cells’ anti-tumor immune response. Furthermore, when NK cells perform an anti-tumor immune function, they do not require the particular procedure of antigen presentation and identification. As a result, NK cell-based immunotherapy in combination with other therapeutic approaches—such as CAR-NK and gene-modified treatment, target blocking, and antibodies—is a viable future path.

Nevertheless, it is impossible to overlook the limitations of even NK cell-based immunotherapy, and the addition of nanomaterials can strengthen the anti-tumor effect of immune cells by enhancing their complementary actions. Additionally, nanomaterials can have their size, surface charge, and shape altered by design. Various kinds of nanomaterials have varying anti-tumor properties. As a result, nanomaterials will play a crucial role in boosting the effectiveness of NK cell tumor immunotherapy. There are nanomaterials that can themselves lessen the inflammatory response of the tumor microenvironment, such as those that encourage immune cells to home in and infiltrate tumor cells. Furthermore, these nanomaterials can direct natural killer cells (NK cells) to eradicate tumor cells and track the real-time efficacy of treatment. It also has an effect on medical picture contrast. As a result, nanomaterials perform a number of roles in the interaction between immune cells and tumor cells, greatly improve NK cell immunotherapy’s anti-tumor efficacy, and open up new application opportunities for tumor immunotherapy.

However, there are certain difficulties in using nanomaterials in therapeutic settings. The primary one is their toxicity and safety. Hepatotoxicity has been linked to metal nanoparticles [84,85,86]. Research has indicated that whilst nanomaterials rely on the liver, chemotherapeutic medicines primarily depend on the kidneys for their breakdown. Nanoparticles build up in tissues close to the liver during liver detoxification, interfering with other enzymes’ ability to break down those tissues [87]. Furthermore, oxidative stress and ferritin precipitation brought on by iron nanoparticles can impact immune cell function and clinical treatment outcomes. Therefore, it is important to take into account the toxicity of the nanomaterials themselves and assess their potential cytotoxicity when creating nanomaterials to support NK cells in tumor immune responses. It is therefore extremely promising to build novel nanomaterials that will facilitate the infiltration of endogenous NK cells in tumors and sustain their activation and function over an extended period of time. The efficiency of immunotherapy can also be increased by leveraging the convergence of several fields and employing various design techniques to create immune regulation medications based on nanoparticles.

## Figures and Tables

**Figure 1 biology-13-00153-f001:**
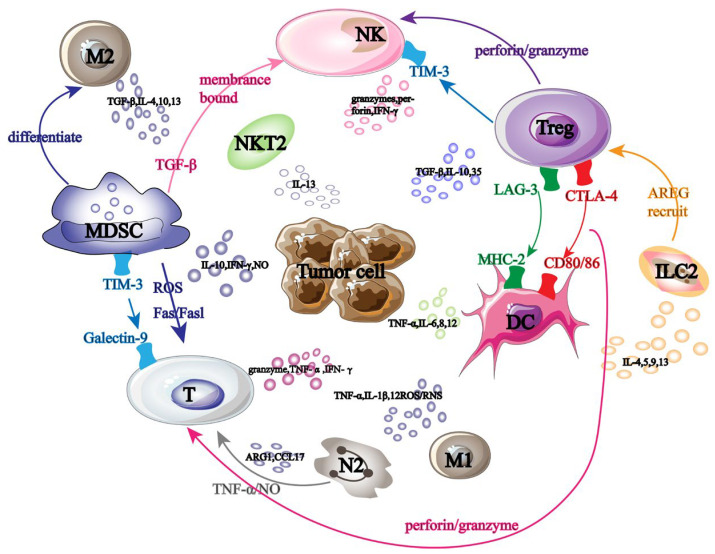
Tumor immunotherapy-related cells. Black large font: names of different tumor-related immune cells; Black font: names of immune factors secreted by tumor-related immune cells; Arrow: the interaction effect between immune cells; Same color font: immune cell surface receptors and ligands.

**Figure 2 biology-13-00153-f002:**
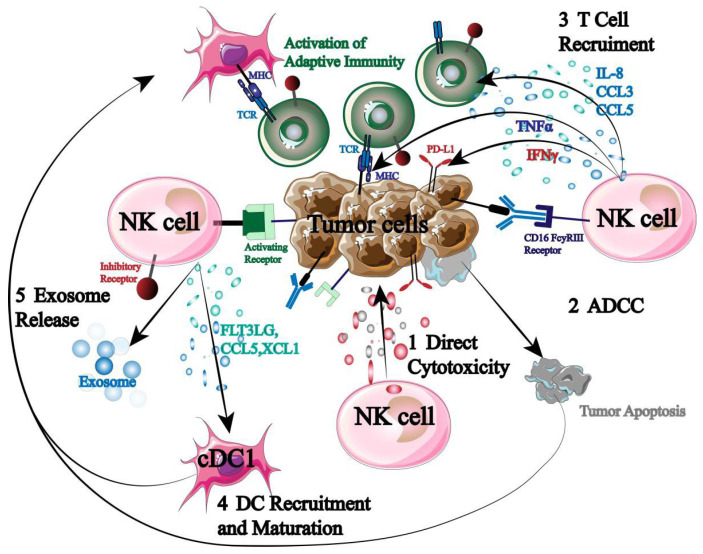
Mechanism of NK cells in tumor immunotherapy.

**Figure 3 biology-13-00153-f003:**
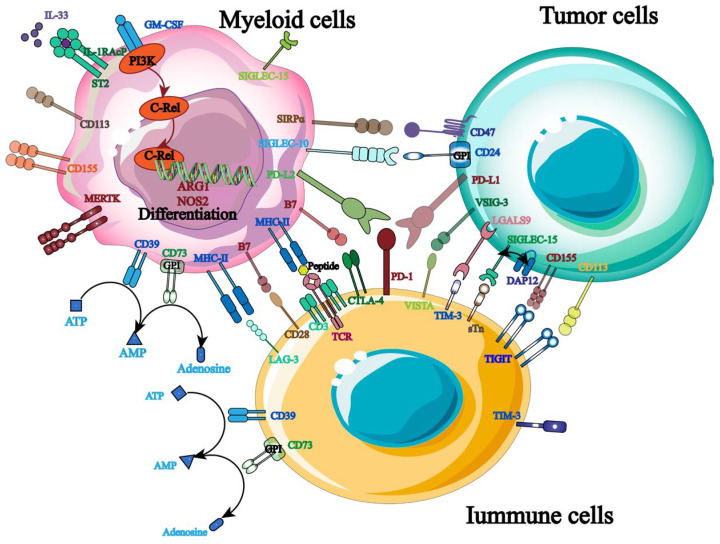
Therapeutic targets related to tumor immunotherapy.

**Figure 4 biology-13-00153-f004:**
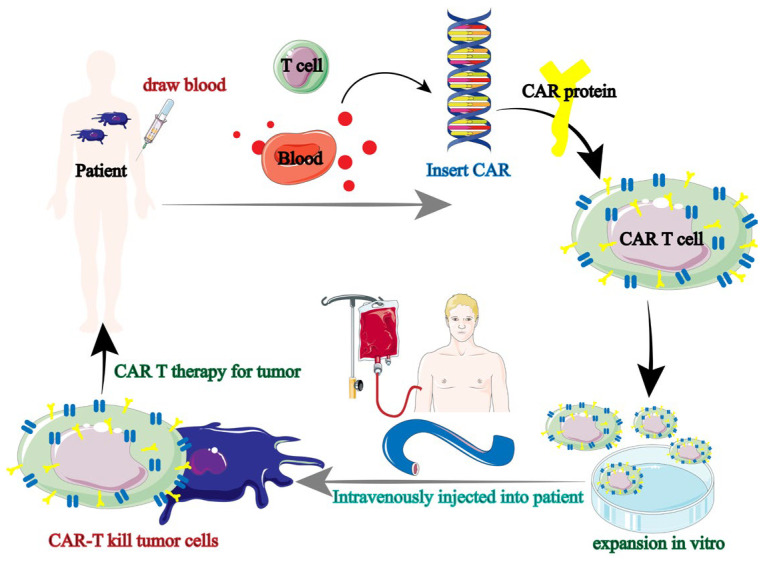
Tumor immunotherapy flow of CAR-T [40].

**Figure 5 biology-13-00153-f005:**
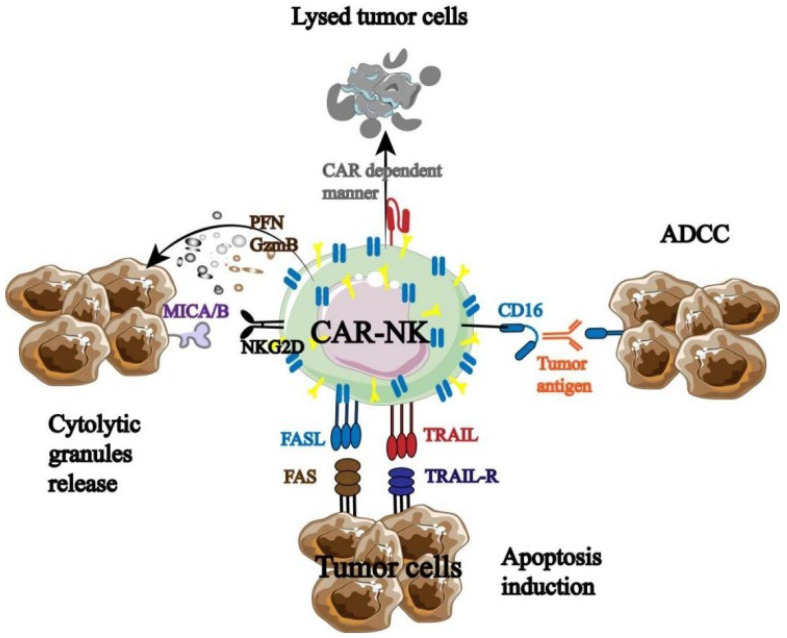
Anti-tumor mechanism of CAR-NK [42].

**Figure 6 biology-13-00153-f006:**
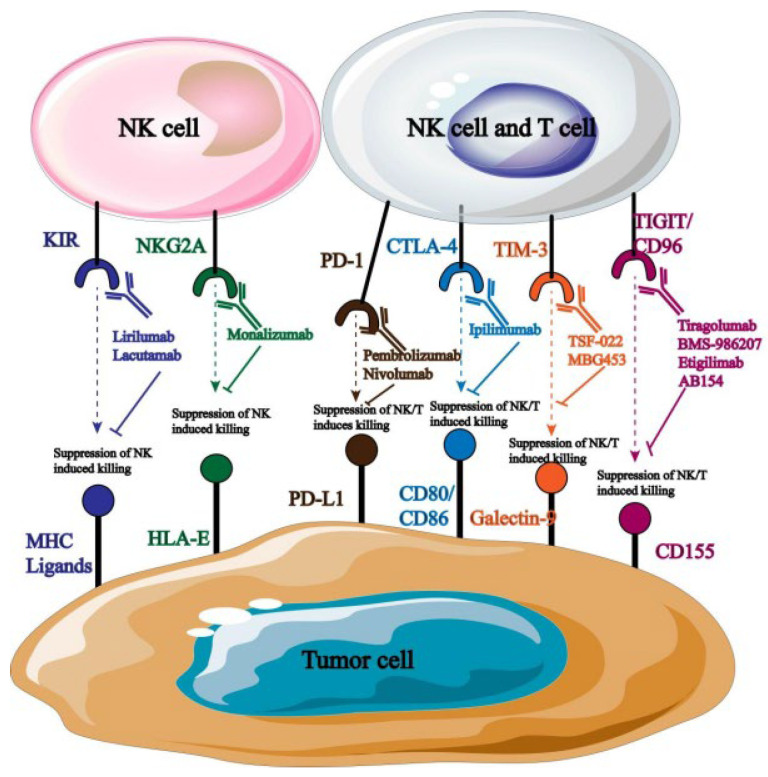
Monoclonal antibodies targeting NK cells.

**Figure 7 biology-13-00153-f007:**
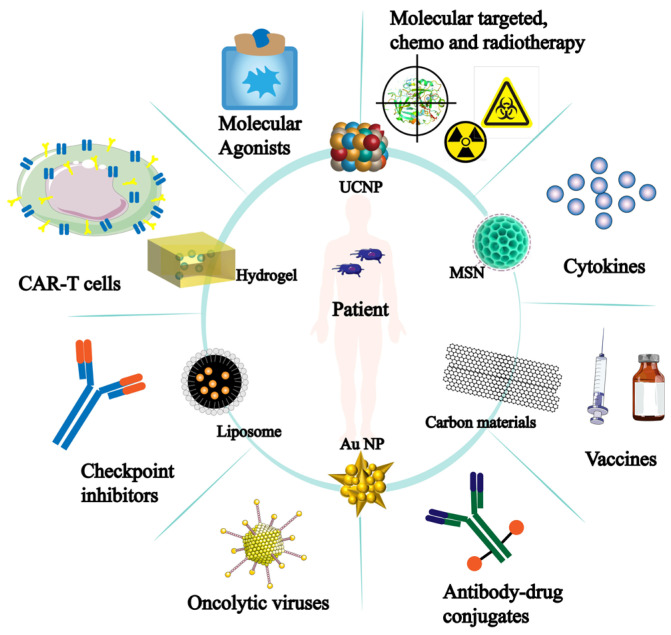
Nanomaterials for tumor immunotherapy. The inner circle represents nanomaterials applied for tumor treatment; The outer circle indicates the effectiveness or direction of these nanomaterials in tumor immunotherapy.

**Table 1 biology-13-00153-t001:** Different sources of NK cells and their own advantages and disadvantages [14].

Sources of NK Cells	Advantages	Disadvantages
Peripheral blood	Safe; conveniently collected; strong ability to kill tumor cells	Low numbers in patients; time-consuming and costly
Umbilical cord blood	Available; off-the-shelf; UCB-derived CD34^+^ cells have been translated to the clinic; frozen for a long time	Only one time to get access to the umbilical cord blood
Human embryonic stem cells or induced pluripotent cell	Homogenous NK cell product; easy to amplify large numbers of NK cells	Need to induce iPSCs into NK cells
Bone marrow	From patients	Invasive operation
NK cell lines (NK-92 and NK-92MI)	Off-the-shelf; easy to amplify; lack most inhibitory receptors compared to naive NK cells	Potential tumorigenicity

## Data Availability

Data are contained within the article.

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
