# Peer review of "Research Progress of Nanomaterials Acting on NK Cells in Tumor Immunotherapy and Imaging"

_biology, 2024, doi:10.3390/biology13030153_

Round 1

Reviewer 1 Report

Comments and Suggestions for Authors

The manuscript of Yachan Feng et al. is devoted to a description of modern approaches to antitumor immunotherapy using various types of cells of the immune system, i.e. immune cell-based anticancer therapy. Along with a detailed description of the features of these approaches, the authors of this literature review discuss advantages and disadvantages of these methods. The main focus of the manuscript is on analyzing the effectiveness of using NK cells and nanomaterials in tumor immunotherapy. It is importent for this review that the authors consider various protocols for the use of these cells in combination with nanoparticles for antitumor therapy, including options for using a number of relevant genetic and other modifications of the immune cells. The authors of this review consider in detail and summarize the published data on the topic of the manuscript. The presented summary of literature data and their analysis give an objective integral picture of existing ideas on the indicated topic and highlight possible ways of practical use of the accumulated knowledge.

In general the review is well presented; however, I have some minor comments:

1. First of all, text of the manuscript contains incorrect and obscure phrases, for example:

(line 2) “   NK Cell-based Nanomaterials   “ means nanoparticles created using NK cells;

(line 26, 27) “   Cancer … has seriously hindered the healthy development of human body   “ - very strange statement.

2. The figures in the manuscript do not have generally accepted detailed legends; only their names are given.

3. (line 139) “   to mediate …” probably “to inhibit”, because “    TIGIT also acts as an inhibitory receptor to transmit immunosuppressive signals…” (line 260).

4. (line 145, 146) “   Immune cell-based therapies are mainly called chimeric antigen receptor immunotherapy (CAR),   “ - Immune cell-based anticancer therapy began to be used long before the appearance of CAR-modified immune cells.

5. In Section 3 and in Table 1 authors should mention the approaches to NK cell expansion in vitro. It might improve the overall quality of the manuscript.

6. (line 308) “   the target receptor   “ should be “the inhibitory receptor”.

7. (line 413) “  Engineered T-Cell-Based   “ – Probably “  Engineered T-Cell-Based Applications “.

8. The manuscript contains a number of inaccuracies and misprints in the text.

Reviewer 2 Report

Comments and Suggestions for Authors

This article reviews the recent progress in Natural Killer (NK) cells-based immunotherapy against cancer.

Despite the title and the topic, the review is very lengthy and reports a great deal of basic and textbook general information that is redundant and distracts from the main topic. For example, Section 2 and 4 could be entirely missed without losing any important information or affecting the understanding of the rest of the article.

On the other side, the information reviewed on NK cells is overall very superficial, nearly anecdotal, without an in depth and critical review or for example comparison between studies.

Few examples, not exhaustive:

Section 3.1.3: how is NK cell activity triggered? What signals are important for NK cell activation?

Section 3.2.2: lack of description of CAR receptor in NK cells; what target(s) is/are used so far? Any potential new target? How is the receptor constructed? What intracellular signalling is triggered? These crucial aspects are not discussed. The authors also list some examples of applications without critically reviewing the results. Are these CAR-NK cells efficient? To what extent? What are the hurdles in improving this approach? E.g Line 290:: “the application is inefficient and faces many limitations. ” Please list them and argument.

Section 3.2.3: The authors describe the introduction of a “suicide” gene in CAR-NK cells to reduce toxicities. However they do not mention at all the result of this approach…. Was it is successful? To what extent these toxicities were reduced, if at all?

Line 321: “good research results have been achieved”. Please expand, and critically review them.

Lines 506-7: in what host were these results reported ? Human, mouse? Here and throughout the review a critical comparison between what has been observed so far in animal models and clinical translation is missing.

Comments on the Quality of English Language

English language should be revised.

For example:

Line 129: “autoimmune expression” I think the authors meant “autoimmune reaction”.

Line 182: “against the virus” ? This is clearly wrong as tumors are not viruses…the sentence should be rephrased and clarified

Section 2.2.2: really difficult to read, should be rephrase and expanded

Round 2

Reviewer 2 Report

Comments and Suggestions for Authors

The authors mostly addressed the points previously raised. Although few sections need further details or ameliorations according the comments below:

Line 172: what do the authors mean with “structural therapy” ? not clear….

Lines 195-206: what do the authors mean with “soluble viruses” ? not clear to me… This is not a standard classification, to my knowledge.

Lines 237-8: what does it mean to administer before infusion?....I think the sentence should be rephrased for better clarity

Lines 270-289: reference to key studies reporting these observations are missing throughout this paragraph.

Lines 244-5: Is the safety improved or equal? The sentence is contradictory since it talks about equality and advantages at the same time…..please rephrase

Line 361: what does it mean “limited drug selection” ? Do the authors refer to a paucity of drug targets?..not clear, please rephrase

Lines 376-381: reference missing

Comments on the Quality of English Language

Line 161: “tumor specific” should be “tumor-specific”

Line 162: “and eradicate THE tumor cells” please remove “THE”

Line 234: “”for cell expansion” should be removed; redundant petition.

Line 235: “the desired phenotype ARE obtained” should be “IS”
